# Selective catalytic two-step process for ethylene glycol from carbon monoxide

Kaiwu Dong[1], Saravanakumar Elangovan[1], Rui Sang[1], Anke Spannenberg[1], Ralf Jackstell[1], Kathrin Junge[1], Yuehui Li[1,2] & Matthias Beller[1]

Upgrading C1 chemicals (for example, CO, CO/$H_2$, MeOH and $CO_2$) with C–C bond formation is essential for the synthesis of bulk chemicals. In general, these industrially important processes (for example, Fischer Tropsch) proceed at drastic reaction conditions ($>250\,°C$; high pressure) and suffer from low selectivity, which makes high capital investment necessary and requires additional purifications. Here, a different strategy for the preparation of ethylene glycol (EG) via initial oxidative coupling and subsequent reduction is presented. Separating coupling and reduction steps allows for a completely selective formation of EG (99%) from CO. This two-step catalytic procedure makes use of a Pd-catalysed oxycarbonylation of amines to oxamides at room temperature (RT) and subsequent Ru- or Fe-catalysed hydrogenation to EG. Notably, in the first step the required amines can be efficiently reused. The presented stepwise oxamide-mediated coupling provides the basis for a new strategy for selective upgrading of C1 chemicals.

[1] Leibniz-Institut für Katalyse e.V. an der Universität Rostock, Albert-Einstein Straße 29a, 18059 Rostock, Germany. [2] State Key Laboratory for Oxo Synthesis and Selective Oxidation, Suzhou Research Institute of LICP, Lanzhou Institute of Chemical Physics (LICP), Chinese Academy of Sciences, 730000 Lanzhou, P.R. China. Correspondence and requests for materials should be addressed to Y.L. (email: yuehui.li@catalysis.de and yhli@licp.cas.cn) or to M.B. (email: matthias.beller@catalysis.de).

A major source for innovation in chemistry and material sciences is the use of easily available feedstocks in new transformations. Owing to the rising global demand for bulk chemicals and materials, there exists strong research interest to discover novel production processes from alternative resources[1]. As central raw materials for the chemical industry, C1 compounds such as CO, $CH_3OH$, $CO_2$ and HCN can be obtained on large scale including renewables. Specifically, CO, which is a most versatile and cost-efficient chemical building block, is easily produced from natural gas, coal, biomass and so on, and is currently used in the industrial production of methanol, higher alcohols, aldehydes, alkanes and diols[2–4]. Despite all these processes, selective coupling of CO still represents a major challenge in C1 chemistry. Although numerous selective C–C coupling reactions using CO are known with olefins, alkynes and C–X bonds, the selective dimerization and oligomerization of CO are basically unknown, despite the huge commercial interest in this area. The problem to efficiently create C–C bonds via coupling of carbon monoxide relies on the high activation barrier for reductive transformations. Hence, despite the apparent simplicity there exists no direct method to convert CO selectively into valuable C2 chemicals[5]. Among these products, ethylene glycol (EG) is an important bulk chemical with an annual production and consumption of more than 20 million tons[6]. As a prevalent industrial product, this C2 diol finds widespread applications in different fields, for example, as a solvent, anti-freeze agent and precursor for the manufacture of polyester fibres, resins and so on. In fact, the growing demand for PET (polyethylene terephthalate) resins and polyester fibres leads to a constant growth of the global market for high-priced fibre-quality EG, which is valued at $33.1 billion in 2014 and is estimated to reach $46.8 billion by 2019 (ref. 7).

Owing to the problems of CO dimerization, today's main technology for large-scale industrial production of EG still relies on the vapour-phase oxidation of ethylene to ethylene oxide started by Carbide Company in the 1930s (refs 8,9). Compared with ethylene, CO is easily available from renewable feedstock, for example, biogas. In addition, it can be considered economically advantageous, although the price for such bulk feedstocks varies quite significantly (for example, from 2000 to 2012 ethylene prices were in between $400 and $1,800 per ton and around $750 per ton in early 2015; see http://www.duncanseddon.com/images/ethylene-price-trends.gif.). Therefore, an increasing interest in industry and academia exists to use CO for the production of EG. However, known methods suffer from the necessity of harsh conditions and the insufficient selectivity. Specifically, this latter problem is a critical issue for EG production. For example, fibre-grade EG (ca. 40% of production capacity in the world) tolerates only small amounts of glycol oligomers, which constitute major byproducts in the conventional route via ethylene oxide hydrolysis.

As discussed above, the direct synthesis of EG starting from syngas under reductive conditions represents an ideal method, but faces significant difficulties to overcome the low reactivity and selectivity. Thus, alternative methods are under development. Accordingly, EG preparation from methanol, formaldehyde and methyl formate are notable, but low selectivity limits their application too[10–13]. Contrary to all these reductive coupling strategies, oxidative dimerization of CO to oxalates might allow for improved efficiency and two main approaches were developed since the 1990s as follows: (1) carbonylation of MeOH with oxygen in the presence of a Pd/V/Ti system[14] and 2) nitric oxide-mediated carbonylation of alcohols to dialkyl oxalates using palladium complexes. Owing to the higher reactivity, the latter system is preferred in industry and has been applied on ton scale[15]. However, the sensitivity of the oxalate intermediate to

water and the use of highly corrosive nitric oxide complicate the latter process and require specific quality of starting materials. Therefore, the development of a selective EG production process from CO without the use/formation of unstable reagents still constitutes a major challenge for industry and continues to attract intense interests of both academic and industrial chemists.

Based on our interest in the catalytic hydrogenation of carboxylic acid derivatives[16], we got inspired to use oxamides as key intermediates for a novel EG process. Notably, the key C–C bond formation in this process should be facilitated in the presence of amines compared with alcohols or water[17]. As shown in Fig. 1, this concept is based on oxidative coupling of CO to oxamides and subsequent selective reduction to EG (to the best of our knowledge, such a reaction has been only proposed in a patent by Shell. However, the selectivity for EG was only ca. 65%. For details, see[18]). Although the overall transformation looks apparently simple, there exist several challenges for the individual transformations. For successful oxidative carbonylation of CO to oxamides, side reactions such as oxidation of CO to $CO_2$ and the formylation of amines to formamides or oxycarbonylation of amines to ureas are well known vide infra and have to be avoided. In addition, decomposition of the active oxidation species in the presence of carbon monoxide has to be carefully controlled. Obviously, the formation of robust catalysts is critical for this oxidative reaction. The other major challenge of this process is how to achieve selective hydrogenation of oxamides to EG, which, to the best of our knowledge, has not been reported yet. Problematic for this transformation is the deactivation effect that might be caused by coordination of the metal centre with the chelating carbonyl groups of the substrate or by product inhibition. Finally, the strong reduction ability of the formed 1, 2-diol facilitates catalyst deactivation.

Despite all these problems, here we disclose that EG can be selectively produced (up to 99% for CO) by sequential Pd-catalysed oxidative carbonylation of piperidine to the corresponding oxamide and subsequent Ru- or Fe-catalysed hydrogenation to EG. The free amine is easily separated from the reaction mixture and recycled.

## Results

**Amines to oxamides**. Although reactions with CO constitute powerful tools for the introduction of carbonyl groups into all kinds of organic molecules[19–21], oxidative carbonylations including the synthesis of heterocycles, carbonates, carbamates, ureas and oxamides are still challenging regarding catalyst productivity and selectivity[22,23]. For example, reactions of amines with CO in the presence of an oxidant lead typically to mixtures with the corresponding urea as the main product[24]. Hence, even though the synthesis of oxamides represents a valid method to couple two CO molecules, it only attracted limited interest for decades. In fact, the state-of-the-art methodology

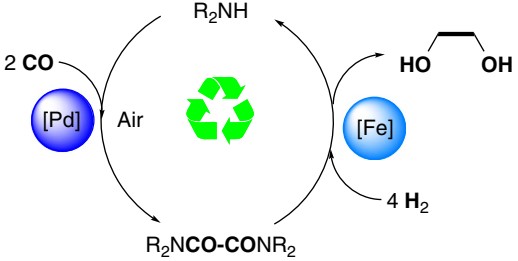

**Figure 1 | Coupling of CO to EG.** Schematic pathway for two-step preparation of EG from CO.

**Table 1 | Towards a practical oxidative carbonylation protocol: influence of ligands and comparison with previously reported catalysts\*.**

| Entry | Pd source | Ligand | [Ox] | Yield (%) | TON | TOF (h$^{-1}$) | Selectivity 2 (%) |
|---|---|---|---|---|---|---|---|
| 1 (ref. 25) | Pd(OAc)$_2$ | — | O$_2$ | 38 | 2.8 | 1 | — |
| 2 (ref. 25) | Pd(OAc)$_2$ | PPh$_3$ | O$_2$ | 88 | 6.6 | 2 | — |
| 3 | Pd(acac)$_2$ | 2, 2'-bipyridine | air | 85 | 42 | 14 | 99 |
| 4 | Pd(acac)$_2$ | H$_2$IMes-CO$_2$ | air | 28 | 14 | 5 | 99 |
| 5 | Pd(acac)$_2$ | P($p$-tol)$_3$ | air | 47 | 24 | 8 | >99 |
| 6 | Pd(acac)$_2$ | Xantphos | air | 99 | 50 | 17 | >99 |
| 7 | Pd(acac)$_2$ | P($o$-tol)$_3$ | air | 99 | 50 | 17 | >99 |
| 8$^†$ | Pd(acac)$_2$ | P($o$-tol)$_3$ | air | 53 | 26,500 | 158 | 99 |
| 9$^‡$ | Pd(acac)$_2$ | P($o$-tol)$_3$ | air | 60 | 3,000 | 750 | 90 |

acac, acetylacetonate; H$_2$IMes-CO$_2$, 1, 3-dimesityl-4, 5-dihydro-1$H$-imidazol-3-ium-2-carboxylate; TOF, turnover frequency; TON, turnover number; Xantphos, (9, 9-dimethyl-9$H$-xanthene-4, 5-diyl)bis(diphenylphosphane).
\*Reaction conditions for ref. 25: 1.5 mmol **1**, 0.1 mmol Pd(OAc)$_2$, 0.2 mmol K$_2$CO$_3$, 0.2 mmol $^n$Bu$_4$NI, 2.7 bar CO and 1 bar O$_2$. For this work: 1.0 mmol **1**, 1 mol% Pd(acac)$_2$ (to **1**), 2 mol% monodentate ligand or 1 mol% bidentate ligand, 10 mol% K$_2$CO$_3$, 5 mol% $^n$Bu$_4$NI, 25 bar CO and 25 bar air. TON = mol of **2** obtained per mol of Pd employed. TOF = mol of **2** obtained per mol of Pd employed per hour.
†0.001 mol% catalyst, 168 h.
‡0.01 mol% catalyst, 120 °C, 4 h.

for this reaction has been disclosed by Pri-Bar and Alper[25], who reported the Pd-catalysed oxycarbonylation of amines to oxamides in the presence of iodide anions as activator. Although the reactions proceeded well under mild conditions using O$_2$ as oxidant, substantial amounts of Pd catalyst (6.7 mol%) are needed to obtain the desired product in high yields (Table 1, entries 1–2).

To bring this reaction closer to practical application, considerable improvements in catalyst efficiency are required (see Supplementary Tables 1–7 for details). To achieve this goal, we envisioned the use of alternative ligands, which should better stabilize the active metal centre. For initial investigations of the ligand effect, the carbonylation of piperidine to 1, 1'-oxalyl dipiperidine (**2**) was performed with air (25 bar) at RT in the presence of various of palladium salts. Indeed, in the presence of nitrogen-based ligands, which are used routinely in palladium-catalysed oxidations, a slight increase in catalyst productivity was observed (Table 1, entry 3). Nevertheless, using nitrogen ligands such as bipyridine did not lead to a stable system and a dark-coloured suspension was formed after the reaction, indicating the decomposition of the active homogeneous catalyst. On the other hand, the use of stronger donating carbene ligand prevented the formation of palladium black and resulted in a minor improvement (Table 1, entry 4). Testing different mono- and bidentate P-based ligands revealed in general a decrease of activity for bidentate phosphines (Table 1, entries 5–6). However, to our surprise, the best result was obtained when using tri($o$-tolyl)phosphine in combination with Pd(acac)$_2$ as the metal salt (Table 1, entry 7). Here, a colourless suspension was obtained after the reaction, indicating the efficient stabilization effect by this ligand. Interestingly, the corresponding palladacycle showed no reactivity at all, which excludes this species as an active intermediate[26]. Other ligands tested gave only inferior results compared with P($o$-tol)$_3$ (see Supplementary Table 2 for details).

Obviously, in the absence of O$_2$, this reaction does not occur. Furthermore, no reactivity was observed when using Ni- or Co-based salts under similar conditions. Using 1 mol% of the optimized palladium catalyst, full conversion of piperidine is observed (>99% yield of **2**), demonstrating that the amine is not the limiting reagent under these conditions. Notably, decreasing the metal loading to 0.001 mol% at RT and similar $P_{CO}$ and $P_{air}$ gave 26,500 catalytic turnovers and no byproducts such as ureas or imines were detected (Table 1, entry 8), which demonstrates that this palladium complex is one of the most efficient catalysts for oxidative carbonylations known to date. As shown in Table 1, when reducing the catalyst loading by a factor of 1,000, the activity of the catalyst system (turnover frequency) increased from 17 to 158 h$^{-1}$ (Table 1, entry 7 versus 8). This surprising increase of activity is explained by the lower conversion in the latter case.

Compared with PPh$_3$ as the ligand, the addition of P($o$-tol)$_3$ improved the stability and efficiency of the corresponding Pd complex under similar conditions. In fact, 60% of PPh$_3$ was oxidized to P(O)Ph$_3$ after 3 h (CO/air = 25/25 bar at RT), whereas < 20% of P($o$-tol)$_3$ was transformed into the corresponding phosphine oxide after the same time. Notably, at low catalyst loading (0.001 mol% Pd/1.2 mol% P($o$-tol)$_3$), even after 7 days around 50% of the free phosphine was present as analysed by gas chromatography. Furthermore, the stability of the catalyst system is shown by three consecutive runs on 10-g scale and 30.9 g of **2** was obtained conveniently after crystallization (see Supplementary Methods for details). In addition, this reaction can be performed at higher temperature (up to 120 °C) and the reactivity increased with a turnover frequency up to 750 h$^{-1}$ (Table 1, entry 9). It is important to note that CO is highly selectively converted to oxamide and no CO$_2$ was detected by gas chromatography in the gas phase after the reaction.

**Hydrogenation of oxamides.** Having an efficient procedure for selective preparation of oxamides in hand, we explored its hydrogenation to EG in the presence of Ru catalysts containing hydride phosphine-amine or pincer ligands. Recently, such organometallic complexes were reported for the reduction of carboxylic acid derivatives[27,28]. As an example, John and Bergens[29] described ruthenium complexes with aminophosphine ligands as one of the most efficient homogeneous catalysts for hydrogenation of β-lactams.

**Table 2 | Transition metal catalysed hydrogenation of oxamide 2\*.**

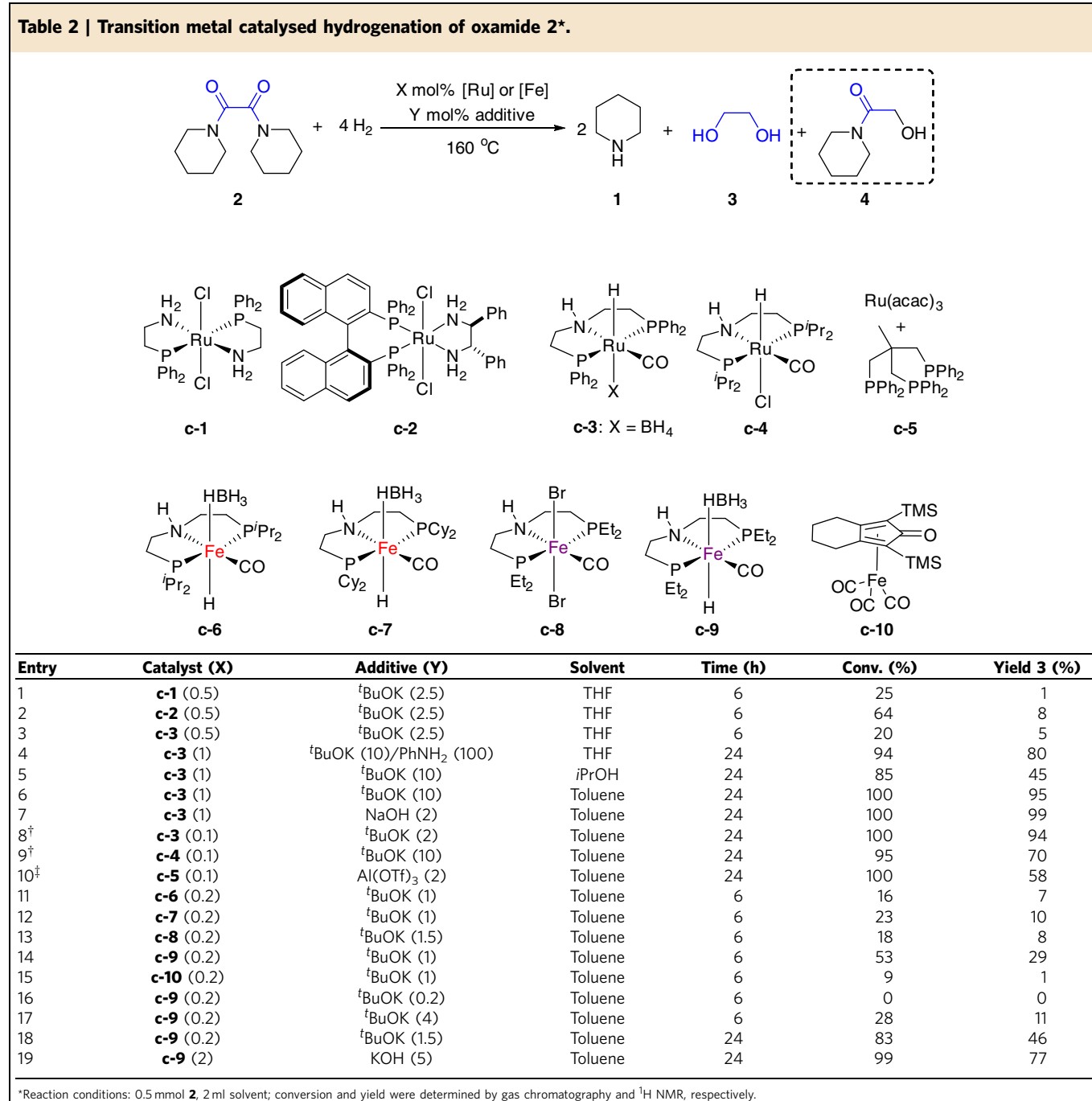

| Entry | Catalyst (X) | Additive (Y) | Solvent | Time (h) | Conv. (%) | Yield 3 (%) |
|---|---|---|---|---|---|---|
| 1 | **c-1** (0.5) | $^{t}$BuOK (2.5) | THF | 6 | 25 | 1 |
| 2 | **c-2** (0.5) | $^{t}$BuOK (2.5) | THF | 6 | 64 | 8 |
| 3 | **c-3** (0.5) | $^{t}$BuOK (2.5) | THF | 6 | 20 | 5 |
| 4 | **c-3** (1) | $^{t}$BuOK (10)/PhNH$_2$ (100) | THF | 24 | 94 | 80 |
| 5 | **c-3** (1) | $^{t}$BuOK (10) | iPrOH | 24 | 85 | 45 |
| 6 | **c-3** (1) | $^{t}$BuOK (10) | Toluene | 24 | 100 | 95 |
| 7 | **c-3** (1) | NaOH (2) | Toluene | 24 | 100 | 99 |
| 8[†] | **c-3** (0.1) | $^{t}$BuOK (2) | Toluene | 24 | 100 | 94 |
| 9[†] | **c-4** (0.1) | $^{t}$BuOK (10) | Toluene | 24 | 95 | 70 |
| 10[‡] | **c-5** (0.1) | Al(OTf)$_3$ (2) | Toluene | 24 | 100 | 58 |
| 11 | **c-6** (0.2) | $^{t}$BuOK (1) | Toluene | 6 | 16 | 7 |
| 12 | **c-7** (0.2) | $^{t}$BuOK (1) | Toluene | 6 | 23 | 10 |
| 13 | **c-8** (0.2) | $^{t}$BuOK (1.5) | Toluene | 6 | 18 | 8 |
| 14 | **c-9** (0.2) | $^{t}$BuOK (1) | Toluene | 6 | 53 | 29 |
| 15 | **c-10** (0.2) | $^{t}$BuOK (1) | Toluene | 6 | 9 | 1 |
| 16 | **c-9** (0.2) | $^{t}$BuOK (0.2) | Toluene | 6 | 0 | 0 |
| 17 | **c-9** (0.2) | $^{t}$BuOK (4) | Toluene | 6 | 28 | 11 |
| 18 | **c-9** (0.2) | $^{t}$BuOK (1.5) | Toluene | 24 | 83 | 46 |
| 19 | **c-9** (2) | KOH (5) | Toluene | 24 | 99 | 77 |

\*Reaction conditions: 0.5 mmol **2**, 2 ml solvent; conversion and yield were determined by gas chromatography and $^1$H NMR, respectively.
†1 mmol **2**, 4 ml toluene.
‡0.1 mol% Ru(acac)$_3$, 0.2 mol% triphos (1,1,1-tri(diphenylphosphinomethyl)ethane), 2 ml toluene and 2 ml H$_2$O.

Unfortunately, **2** was not an eligible substrate under similar conditions. This indicates that oxamides are more difficult to reduce compared with simple carboxamides. Hence, only low-to-moderate reactivity was observed by using catalysts **c-1**, **c-2** and **c-3** in the presence of $^t$BuOK (Table 2, entries 1–3), which are well established for ester hydrogenations. In general, in these processes the added base will accelerate the formation of an active Ru amido complex, which facilitates the heterolytic cleavage of H$_2$ to generate the corresponding active Ru-H species in the catalytic cycle.

Kinetic investigations on the hydrogenation catalysed by Ru-MACHO-BH (**c-3**) showed a strong inhibition of activity after *ca.* 30% conversion due to the formation of coordinating intermediates such as glycol amide **4**. Indeed, this half reduction

product was isolated in 24% yield using **c-1** (ref. 30). To solve this problem, different additives and solvents were tested. Gratifyingly, a substantial positive solvent effect was observed by replacing tetrahydrofuran (THF) with isopropanol or toluene, which increased the yield of EG from trace to 45% and 95% yield, respectively (Table 2, entries 3 *versus* 5–6). Finally, we observed the hydrogenation of **2** also works well when inexpensive NaOH was used instead of KO$^t$Bu and full conversion to EG was achieved (Table 2, entry 7). Decreasing the catalyst loading to 0.1 mol% still gave 94% yield of EG (Table 2, entry 8). No byproducts such as tertiary amines that might be easily formed via amination of EG were observed. Notably, under these mild conditions, over-reduction to ethanol was not detected by $^1$H NMR measurement of the reaction mixture. Interestingly, the

Ru-triphos system **c-5** gave the desired product in 58% yield in the presence of $Al(OTf)_3$ (ref. 31), whereas other catalysts showed no reactivity at all with acid additives (see Supplementary Tables 8–11 for details).

Owing to the obvious advantages of base metals compared with noble metal complexes, we were interested to investigate Fe catalysts for the hydrogenation of **2**. Based on the recent development of Fe pincer complexes (such as **c-6**) for hydrogenation of esters and nitriles[32–34], we prepared a series of complexes **c-7-9** (see Supplementary Fig. 1 for spectra of **c-8**).

Indeed, selective hydrogenation took place and, similar with the reactions using Ru-catalysts, worse reactivity was obtained in THF compared with the reactions in toluene. Moderate reactivity was obtained with Fe pincer complexes, whereas Knölker's Fe complex only produced trace amounts of product (Table 2, entries 11–15)[35]. The concentration of base was critical to achieve higher yields of EG (Table 2, entries 16–17) and the best reactivity was obtained with **c-9** (Table 2, entry 18). Using 2 mol% of **c-9**, full conversion was achieved with **4** as the only byproduct (Table 2, entry 19).

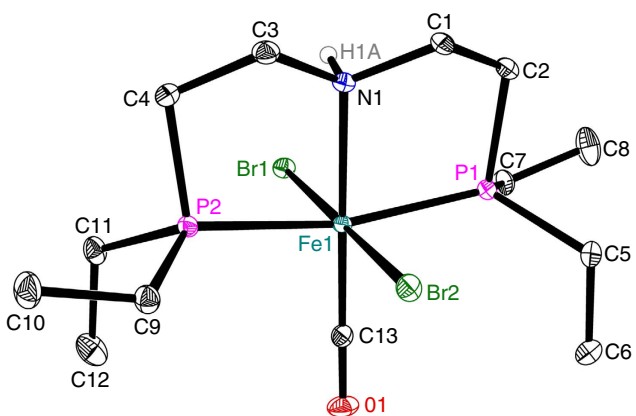

**Figure 2 | Molecular structure of complex c-8.** Displacement ellipsoids are drawn at the 30% probability level. Hydrogen atoms, except of H1A, are omitted for clarity. Selected bond lengths (Å) and angles (°): Fe1-N1 2.0783(10), Fe1-C13 1.7499(13), Fe1-P1 2.26960(4), Fe1-P2 2.2683(4), Fe1-Br1 2.4735(2), Fe1-Br2 2.4562(2), C13-O1 1.1477(16); C13-Fe1-N1 179.05(5), P1-Fe1-P2 167.29 (13), Br1-Fe1-Br2 178.339(8) and N1-Fe1-P1 83.02(3).

The X-ray crystal structure analysis of complex **c-8**, a precursor of **c-9**, reveals a distorted octahedral coordination geometry around the Fe(II) centre, with the CO ligand located *trans* to the nitrogen atom and the two bromine atoms located *trans* with each other (Fig. 2). Apparently, the sterically less demanding $PEt_2$ groups allow for an easier access of the substrate molecule compared with catalyst **c-6**.

Finally, it should be noted that both the iron- and ruthenium-based catalysts can be used for hydrogenation of other oxamides, for example, oxanilide (see Supplementary Tables 12–13 for details), as well as oxalates[36].

## Discussion

The two reaction cycles for the selective synthesis of EG from carbon monoxide are shown in Fig. 3. In the first step, secondary amines such as piperidine or pyrrolidine reacted smoothly to the corresponding oxamides[37–39]. Initially, the piperidine-derived bis(carbamoyl)palladium intermediate[40] is formed from the corresponding monocarbamoyl palladium intermediate at RT highly selectively. Thus, no urea products are detected, which are common side products in such oxidative carbonylations. Next, the corresponding oxamides are formed via reductive elimination giving Pd(0) species. Finally, oxidation with molecular oxygen leads to the regeneration of the active Pd(II) species. Notably, iodide promotes both the generation of the carbamoyl group and the reoxidation of Pd(0)[23]. Besides, the addition of $(o\text{-tol})_3P$ significantly improved the efficiency of the catalyst system despite the well-known sensitivity of phosphine ligands towards oxygen. After the oxidative carbonylation, the solvent (THF) was removed *in vacuo* from the reaction mixture of step 1 followed by addition of toluene. The resulted mixture was quickly filtrated (silica gel, 2–3 cm) and then subjected to the hydrogenation catalyst without further purification (Fig. 4 and see Supplementary Table 14 for details). Subsequent oxamide hydrogenation should proceed via an outer-sphere bifunctional activation mode when using M-$PN^HP$ type Ru complex **c-3** or Fe complex **c-9**. In the presence of these catalysts, no formation of water was observed, which suggests the direct hydrogenolysis of the amide bond of intermediate **b'**. From a practical point of view, it is noteworthy that the reaction system is water tolerant, as control experiments with addition of water showed no influence. Based on the two steps shown in Fig. 4, the preparation of EG from CO proceeds with remarkable selectivity (99%). Final purification of the desired

**Figure 3 | Proposed reaction cycles for piperidine-mediated production of EG from CO.** Step 1: Pd-catalysed oxidative carbonylation of piperidine to oxamide **2**. Step 2: Ru-catalysed hydrogenation of oxamide **2** to EG and piperidine.

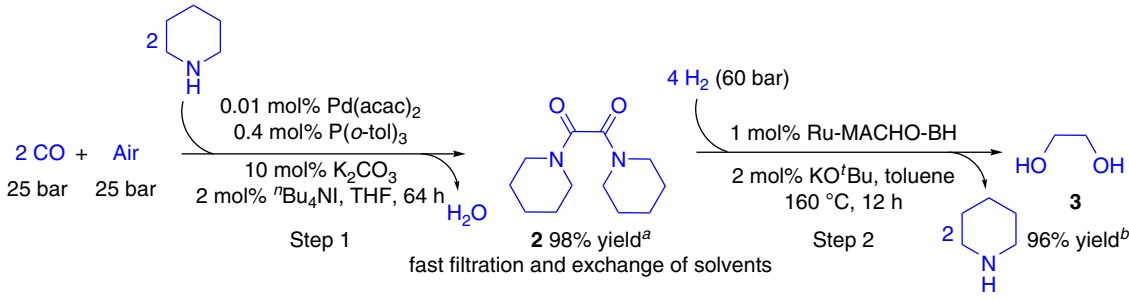

**Figure 4 | Piperidine-mediated production of EG from CO via two steps.** Reaction conditions for step 1: Pd(acac)$_2$ (6.1 mg, 0.02 mmol), P(o-tol)$_3$ (244 mg, 0.8 mmol), $^n$Bu$_4$NI (1.48 g, 4 mmol), K$_2$CO$_3$ (2.76 g, 20 mmol), piperidine (20 ml, 200 mmol, in three portions), THF (100 ml), room temperature (RT), 64 h. For step 2: **2** (5.0 mmol in 12 ml toluene), Ru-MACHO-BH (29 mg, 0.05 mmol), KO$^t$Bu (11.2 mg, 0.1 mmol), toluene (8 ml), H$_2$ (60 bar), 160 °C, 12 h. $^a$Gas chromatography yield using iso-octane as internal standard. $^b$ $^1$H NMR yield using tert-butanol as internal standard.

product from the solvent (toluene) and the co-product (piperidine) is comparably easy due to the formation of a biphasic mixture. Simple phase separation provides EG with a purity of >95%.

In conclusion, we demonstrated for the first time the selective synthesis of EG from CO, H$_2$ and air. By combination of an efficient room-temperature Pd-catalysed oxidative carbonylation with a selective hydrogenation, it is possible to produce C2 compounds with excellent selectivity (>99% based on CO) and good reactivity. Amines were used and recycled without observable consumption. Key to success of this work is the selective construction of C2 oxamide intermediates. Although the presented catalytic efficiencies are already high for academic standards, clearly a bulk industrial implementation requires further optimization. However, we are confident that the presented methodology has a high potential to become an economical viable process for EG production. At the same time, this work will inspire chemists to discover new concepts on selective transformation of C1 chemicals to C2 or even higher chemicals.

**Data availability**. CCDC 1426403 contains the supplementary crystallographic data for this paper. These data can be obtained free of charge from the Cambridge Crystallographic Data Centre via www.ccdc.cam.ac.uk/data_request/cif. All other data are available from the authors upon reasonable request.

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

## Acknowledgements

This work is supported by Chem21, Danish National Research Foundation CADIAC (Carbon Dioxide Activation Centre), the state of Mecklenburg-Vorpommern and the BMBF (Bundesministerium für Bildung und Forschung). We thank Bianca Wendt for her kind support.

## Author contributions

M.B. and Y.L. designed the method. M.B. directed and coordinated the project. Y.L., K.D. and R.S. conducted the experiments and developed the project. S.E. and A.S. prepared and analysed the single crystal of complex **2b**. M.B., Y.L., K.J. and R.J. wrote the manuscript.

## Additional information

**Competing financial interests:** The authors declare no competing financial interests.

