## [Peer review file · Nature Communications]

Reviewers' comments:

Reviewer #1 (Remarks to the Author):

This manuscript describes an indirect strategy to convert carbon monoxide to ethylene glycol. The results are of interest from a fundamental research perspective but have low potential for practical applications. While this work represents a solid advance of the field, it is not at the level to merit publication in Nature.

Reviewer #2 (Remarks to the Author):

Ethylene glycol (EG) is an important bulk chemical with an annual production and consumption more than 20 million tons with widespread applications in different fields, e.g. as solvent, antifreeze agent and precursor for the manufacture of polyester fibers, resins etc. The current processes for industrial production of EG include vapor-phase oxidation of ethylene to ethylene oxide followed by ethylene oxide hydrolysis, and nitric oxide (NO) mediated carbonylation of alcohols to dialkyl oxalates using palladium complexes followed by hydrogenation of the resultant dialkyl oxalates. Both processes actually suffer from various disadvantages. In the present communication, Beller and co-workers demonstrated for the first time an alternative approach to the selective synthesis of EG from CO, H₂ and air. The present process includes the combination of an efficient room-temperature Pd-catalyzed oxidative carbonylation in the presence of amine with a selective hydrogenation of the resultant oxamide, featuring excellent selectivity (> 99% based on CO) and good reactivity. At the same time, the amines were able to be used and recycled without observable consumption. The key to success of this work is the selective construction of C₂ oxamide intermediates, although this is a known but overlooked reaction, since the efficiency of this transformation was significantly enhanced by several orders of magnitude via careful tuning the ligand of Palladium catalyst. Although the catalytic efficiencies have been already high from an academic standards, clearly a bulk industrial implementation requires further optimization. But I believe this work represents a significant breakthrough in the fundamental research of CO transformation to C₂ molecules. This work will encourage the chemists to discover new concepts on selective transformation of C₁ chemicals to C₂ or even higher chemicals.

The manuscript is well organized and the presentation is clear enough for understanding.

1) One reference related to the production of Eg should be cited (Angew Chem. Int. Ed. 2012, 51, 13041-13045).

2) Another concern that I have is the separation of EG product from the amine and recovery of the homogeneous catalyst, due to the fact that in the absence of H₂, the homogeneous catalysts (Ru or Fe) will be able to facilitate the dehydrogenation of ethylene glycol or amines, and accordingly negatively impact the quality of the EG product and recovered amine. Did the authors try the separation experiments to confirm the conclusion of "Amines

were used and recycled without observable consumption" and insure the quality of ethylene glycol?

In summary, this work represents a critical contribution to the C1 chemistry and will definitely have significant impact on the green chemistry of C1. I strongly support its publication in Nature Communication after minor revision.

Reviewer #3 (Remarks to the Author):

The authors describe a two-step process for producing ethylene glycol (EG) from carbon monoxide. This is an intriguing idea and one that might be of practical, as well as scientific, interest. The authors propose that this could be done using two catalysts working in tandem - the first promoting the conversion of CO to oxamide and the second one promoting the conversion of oxamide to EG. This work may be publishable but not in its present form. Reasons for concern are given below.

1. The authors assert that the cost of CO is lower than that of ethylene. This statement is quite doubtful and is made without any references.
2. The authors mention the carbonylation of MeOH with oxygen in the presence of a Pd/V/Ti system. This statement needs to be tied to a reference.
3. A variety of phosphine ligands are used in the presence of air (see Table 1). The authors need to state how stable these ligands are to oxidation.
4. The stoichiometry of the reaction appearing above Table 1 is not correct. Water needs to be added to the right-hand side.
5. Entry 8 of Table 1 indicates that reducing the catalyst loading from 1 mol% to 0.001 mol% increased the number of turnovers by ~ 500 fold all other reaction conditions being equal. This is quite odd and begs explanation.
6. The time over which the reactions in Table 2 were carried out must be stated.
7. In discussing the results presented in Table 2, the authors state that the addition of NaOH (entry 7) was beneficial for obtaining complete conversion of oxamide to EG. The authors need to explain why NaOH is beneficial.
8. In discussing Table 2, entry 8, the authors again note that decreasing the loading of catalyst enabled almost complete conversion of oxamide. This statement needs to be explained.

9. Figure 3 describes how the two individual catalytic processes the authors have used can be combined to achieve the conversion of CO to form EG. While this is the most important part of the story, there are no details given. Without such information one must question whether the proposed scheme actually works. The authors must specify the amounts of catalyst and ligand used, the temperature, the rate at which EG is produced, and the stability of the system. Without such information, which must be included in the Results section, the work is intriguing but certainly not worthy of publication in Nature Communications nor any other chemistry journal.

In summary, the work in its current form is not acceptable for publication. There are simply too many loose ends and the way that the work is reported does not allow one to reproduce it.

Response to the Reviewers' comments:

Reviewer #1 (Remarks to the Author):

This manuscript describes an indirect strategy to convert carbon monoxide to ethylene glycol. The results are of interest from a fundamental research perspective but have low potential for practical applications. While this work represents a solid advance of the field, it is not at the level to merit publication in Nature.

Reviewer #2 (Remarks to the Author):

Ethylene glycol (EG) is an important bulk chemical with an annual production and consumption more than 20 million tons with widespread applications in different fields, e.g. as solvent, antifreeze agent and precursor for the manufacture of polyester fibers, resins etc. The current processes for industrial production of EG include vapor-phase oxidation of ethylene to ethylene oxide followed by ethylene oxide hydrolysis, and nitric oxide (NO) mediated carbonylation of alcohols to dialkyl oxalates using palladium complexes followed by hydrogenation of the resultant dialkyl oxalates. Both processes actually suffer from various disadvantages. In the present communication, Beller and co-workers demonstrated for the first time an alternative approach to the selective synthesis of EG from CO, H₂ and air. The present process includes the combination of an efficient room-temperature Pd-catalyzed oxidative carbonylation in the presence of amine with a selective hydrogenation of the resultant oxamide, featuring excellent selectivity (> 99% based on CO) and good reactivity. At the same time, the amines were able to be used and recycled without observable consumption. The key to success of this work is the selective construction of C2 oxamide intermediates, although this is a known but overlooked reaction, since the

efficiency of this transformation was significantly enhanced by several orders of magnitude via careful tuning the ligand of Palladium catalyst. Although the the catalytic efficiencies have been already high from an academic standards, clearly a bulk industrial implementation requires further optimization. But I believe this work represents a significant breakthrough in the fundamental research of CO transformation to C2 molecules. This work will encourage the chemists to discover new concepts on selective transformation of C1 chemicals to C2 or even higher chemicals.

The manuscript is well organized and the presentation is clear enough for understanding.

1) One reference related to the production of EG should be cited (*Angew Chem. Int. Ed.* **2012**, *51*, 13041-13045).

Response: Thanks! The reference (*Angew Chem. Int. Ed.* **2012**, *51*, 13041-13045) was added as *Ref* (9).

2) Another concern that I have is the separation of EG product from the amine and recovery of the homogeneous catalyst, due to the fact that in the absence of H₂, the homogeneous catalysts (Ru or Fe) will be able to facilitate the dehydrogenation of ethylene glycol or amines, and accordingly negatively impact the quality of the EG product and recovered amine. Did the authors try the separated experiments to confirm the conclusion of "Amines were used and recycled without observable consumption" and insure the quality of ethylene glycol?

Response: The concentration of oxamide (< 0.25 M) in the hydrogenation with the Ru or Fe catalyst is crucial to the production of EG and recovering the amine. After the oxamide hydrogenation, the yield of the corresponding amine (piperidine) was determined by GC and ¹H NMR analysis. Based on suggestions from the Reviewer, one reaction was carried out (details shown below) and piperidine was recycled in 94% isolated yield as the hydrochloride salt form. Under argon atmosphere, 29 mg Ru-MACHO-BH (0.05 mmol), 11.2 mg KO^tBu (0.1 mmol), and 1.12 g oxamide **2** (5 mmol) were added to a 100 mL autoclave. At room temperature the autoclave was flushed with H₂ three times and pressurized with H₂ to 60 bar. The reaction was performed at 160 °C for 12 h. Full conversion of **2** was detected by GC analysis. After 11.0 mL of HCl solution (2 M in Et₂O) was added to the reaction solution, significant amounts of precipitate were observed. Then, the mixture was filtrated and 1.14 g yellowish solid was obtained (94% yield). The composition of the solid was determined by ¹H NMR as the hydrochloride salt of piperidine form (> 95% purity).

In summary, this work represents a critical contribution to the C1 chemistry and will definitely have significant impact on the green chemistry of C1. I strongly support its publication in Nature Communication after minor revision.

We thank the reviewer for his/her kind comments.

Reviewer #3 (Remarks to the Author):

The authors describe a two-step process for producing ethylene glycol (EG) from carbon monoxide. This is an intriguing idea and one that might be of practical, as well as scientific, interest. The authors propose that this could be done using two catalysts working in tandem - the first promoting the conversion of CO to oxamide and the second one promoting the conversion of oxamide to EG. This work may be publishable but not in its present form. Reasons for concern are given below.

1. The authors assert that the cost of CO is lower than that of ethylene. This statement is quite doubtful and is made without any references.

Response: We thank the referee for his comment and agree, it is not easy to make a comparison between ethylene and CO prices. On the one hand the price for such bulk feedstocks varies quite significantly. For example from 2000 to 2012 ethylene prices were in between 400-1800 \$ per ton (around 750 \$/ton in early 2015; see <http://www.duncanseddon.com/images/ethylene-price-trends.gif> and related data on the www). On the other hand, carbon monoxide is not traded on a worldwide basis and typically consumed after on site production. Nevertheless, we believe CO constitutes a valuable alternative feedstock for EG production. Regarding its price, the generation of CO from “cheap” coal is possible. In addition, CO is not only available from fossil based feedstock, but also – in a competitive manner – from renewables, e.g. bio-gas. In order to make this clearer to the reader we changed the sentence accordingly.

2. The authors mention the carbonylation of MeOH with oxygen in the presence of a Pd/V/Ti system. This statement needs to be tied to a reference.

Response: A reference has been added to the statement (“carbonylation of MeOH with oxygen in the presence of a Pd/V/Ti system;¹⁴”).

3. A variety of phosphine ligands are used in the presence of air (see Table 1). The authors need to state how stable these ligands are to oxidation.

Response: In the Pd-catalyzed oxidative carbonylation of piperidine, a variety of ligands has been used and the different stability of these ligands towards oxidation could be traced by GC analysis after the reaction. As expected, sterically hindered phosphine ligands as well as electron-deficient ones are more reluctant to be oxidized by oxygen or air. To make this more clear the following sentence has been added to the discussion on Table 1: “Indeed, comparison with PPh₃ as the ligand, the addition of P(*o*-tol)₃ improved the stability and efficiency of the corresponding Pd complexes under similar conditions (under the CO/air pressure of 25/25 bar at RT, 60% of PPh₃ was oxidized into P(O)PPh₃ after 3 h while 5-50% of P(*o*-tol)₃ was transformed into the corresponding phosphine oxide after 3-168 h analysed by GC).”

4. The stoichiometry of the reaction appearing above Table 1 is not correct. Water needs to be added to the right-hand side.

Response: Thank you, we apologize for the mistake. Water has been added to the right-hand side in the equation above Table 1.

5. Entry 8 of Table 1 indicates that reducing the catalyst loading from 1 mol% to 0.001 mol% increased the number of turnovers by ~ 500 fold all other reaction conditions being equal. This is quite odd and begs explanation.

Response: When the catalyst loading was reduced from 1 mol% to 0.001 mol%, the number of catalytic turnovers increased by ~500 fold. However, as stated in the manuscript the reaction time was increased in this case and the yield of the desired oxamide decreased to 53%. In order to make this clear, we have added a column for oxamide yields in Table 1.

6. The time over which the reactions in Table 2 were carried out must be stated.

Response: We added the requested data in Table 2.

7. In discussing the results presented in Table 2, the authors state that the addition of NaOH (entry 7) was beneficial for obtaining complete conversion of oxamide to EG. The authors need to explain why NaOH is beneficial.

Response: When the solvent was changed from THF to toluene (Table 2), selective reduction of oxamide to EG is achieved in the presence of KO^tBu or NaOH. This observation is in agreement with recent reports on the mechanism of PN^HP-pincer Ru-catalyzed hydrogenations, which proceed via an outer-sphere bifunctional activation. Hence, we added the following sentence to the discussion: "In general, the added base will accelerate the formation of an active Ru amido complex, which facilitates the heterolytic cleavage of H₂ to generate the corresponding Ru-H species in the catalytic cycle."

8. In discussing Table 2, entry 8, the authors again note that decreasing the loading of catalyst enabled almost complete conversion of oxamide. This statement needs to be explained.

Response: Comparison of entries 6 and 8 in Table 2 shows a slight decrease of the product yield. Please note that according to entry 6 (1 mol% **c-3**, 100% conv.) EG was obtained in 95% yield, while decreasing the catalyst loading to 0.1 mol% still gave (not enabled) 94% yield of EG in excellent selectivity (Table 2, entry 8).

9. Figure 3 describes how the two individual catalytic processes the authors have used can be combined to achieve the conversion of CO to form EG. While this is the most important part of the story, there no details given. Without such information one must question whether the proposed scheme actually works. The authors must specify the amounts of catalyst and ligand used, the temperature, the rate at which EG is produced, and the

stability of the system. Without such information, which must be included in the Results section, the work is intriguing but certainly not worthy of publication in Nature Communications nor any other chemistry journal.

Response: We apologize for this misunderstanding and agree with the referee that it is important for reader to understand the combined process. As requested, we have added the reaction conditions in detail (as follows) into the main text and the supporting information. Below you will find the modified Scheme 1 and the detailed procedures.

Scheme 1. Piperidine-mediated production of EG from CO via two steps. *a.* GC yield using *iso*-octane as internal standard. *b.* ¹H NMR yield using *tert*-butanol as internal standard.

Detailed experimental procedures were added in SI.

Step 1: In a 300 mL autoclave, 6.1 mg Pd(acac)₂ (0.02 mmol) and 244 mg P(*o*-tol)₃ (0.8 mmol) were dissolved in 100 mL THF. After the solution was stirred at room temperature for 10 min, 1.48 g ⁿBu₄NI (4 mmol), 2.76 g K₂CO₃ (20 mmol), and 5.0 mL piperidine (50 mmol) were added. The autoclave was pressurized with air (25 bar) and CO (25 bar). After the reaction was stirred at room temperature for 16 h, the pressure was released. 99% yield of the corresponding oxamide **2** was observed by GC. Another portion of 5 mL piperidine **1** (50 mmol) was added into the mixture followed by re-pressurizing air (25 bar) and CO (25 bar). The reaction mixture was stirred at room temperature for 16 h. Similarly, two more portions of 5 mL piperidine **1** (50 mmol) were added to the same reactor after 16 h. Altogether, the whole procedure was done corresponding to 0.01 mol% of palladium catalyst loading. After all, 98% yield of the desired product was detected by GC analysis using *iso*octane as the internal standard.

Step 2: THF was removed from 60 mL reaction mixture of **step 1** followed by addition of 120 mL toluene. After stirring at room temperature for 20 min, the resulted mixture was filtrated through thin silica gel (2-3 cm). 5 mL toluene was used to wash the residue. The concentration of oxamide **2** in toluene was determined as follows: 1 mL filtrate was taken out followed by addition of 100 μ L *iso*octane (internal standard), 94 mg oxamide **2** (0.42 mmol) in 1 mL toluene was observed by GC analysis. Then 12 mL toluene solution of **2** (5 mmol) was added into a 100 mL autoclave which charged with 29 mg Ru-MACHO-BH (0.05

mmol) and 11.2 mg KO^tBu (0.1 mmol) under argon atmosphere. 8 mL toluene was injected by syringe to the autoclave. At room temperature the autoclave was flushed with H₂ three times, and then pressurized with H₂ to 60 bar. The reaction was performed at 160 °C for 12 h. Full conversion of **2** was detected by GC analysis. 96% yield of ethyl glycol was measured by ¹H NMR analysis using *tert*-butanol as internal standard.

The following Table and Figure have been added to the SI.

Table S15. Variation of reaction conditions in step 2 using the reaction mixture of step 1 as the starting material.

Discussion: Notably, the hydrogenation of **2** in toluene is more efficient than in THF when using isolated **2** as the reactant. Therefore, we investigated the solvent effect with the reaction mixture of step 1 as the reactant (entry 1: THF, entry 2: THF/toluene, entry 3: toluene). Unfortunately, in all cases only low conversion of **2** was detected (5-28%). However, after short filtration (2-3 cm silica gel) to remove undissolved Pd particles EG was obtained in 98% yield as detected by ¹H NMR. When increasing the amount of the reaction mixture of step 1 from 6 mL to 20 mL, the reaction gave 92% yield of EG (entry 5).

Entry	Procedure for "one-pot" production of EG	Conv.(%) ^a	Yield/% ^b
1	After hydrogenation, 20 mL THF was added to 6 mL of the THF reaction mixture of step 1. Then, the newly resulting mixture was added to a 100 mL autoclave which was charged with 58 mg Ru-MACHO and 22.4 mg KO ^t Bu. After the autoclave was flushed with H ₂ three times and pressurized with H ₂ to 50 bar, the reaction was performed at 160 °C for 20 h.	26	--
2	After hydrogenation, 20 mL toluene was added to 6 mL of the THF reaction mixture of step 1. Then, the resulting mixture was added to a 100 mL autoclave which was charged with 58 mg Ru-MACHO and 22.4 mg KO ^t Bu. After the autoclave was flushed with H ₂ three times and pressurized with H ₂ to 50 bar, the reaction was performed at 160 °C for 20 h.	28	--

3	After hydrogenation, the solvent was removed in vacuo from 6 mL of reaction mixture of step 1 followed by addition of 20 mL toluene. The resulting mixture was stirred at RT for 20 min and then added to a 100 mL autoclave which was charged with 58 mg Ru-MACHO and 22.4 mg KO ^t Bu. After the autoclave was flushed with H ₂ three times and pressurized with H ₂ to 50 bar, the reaction was performed at 160 °C for 20 h.	5	--
4	After hydrogenation, THF was removed in vacuo from 6 mL of the reaction mixture of step 1 followed by addition of 20 mL toluene. The resulting mixture was stirred at RT for 20 min and then filtrated (silica gel, 2-3 cm). The filtrate was injected by syringe to a 100 mL autoclave which was charged with 58 mg Ru-MACHO and 22.4 mg KO ^t Bu. After the autoclave was flushed with H ₂ three times and pressurized with H ₂ to 50 bar, the reaction was performed at 160 °C for 20 h.	>99	98
5	After hydrogenation, THF was removed from 20 mL of the reaction mixture of step 1 followed by addition of 20 mL toluene. The resulting mixture was stirred at RT for 20 min and then filtrated through thin silica gel (2-3 cm). The filtrate was injected by syringe to a 100 mL autoclave which was charged with 58 mg Ru-MACHO and 22.4 mg KO ^t Bu. After the autoclave was flushed with H ₂ three times and pressurized with H ₂ to 50 bar, the reaction was performed at 160 °C for 20 h.	>99	92

a. Determined by GC using *iso*-octane as internal standard. *b.* ¹H NMR yield using *tert*-butanol as internal standard.

Figure S1. H₂ pressure profile for Entry 5, Table S15. When the reaction temperature increased from RT to 160 °C, the H₂ pressure increased from 51 bar to 63 bar. A significantly gas consumption (about 10 bar) was observed after the reaction was carried out at 160 °C for 3 h.

REVIEWERS' COMMENTS:

Reviewer #2 (Remarks to the Author):

My concern has been well addressed by authors in the revised manuscript. It is now ready for acceptance.

Reviewer #3 (Remarks to the Author):

The authors have addressed most of my questions but not all. The remaining issues are listed using the same numbering scheme as that used in the original review.

1. The authors needs to document with numbers that the cost of synthesis gas from methane or coal is lower than the price of ethylene. Simply expressing the belief is not sufficient.

5. The authors need to give a response for why lowering the amount of catalyst by a factor of 1000 increased the number of turnovers by a factor of 500. Turnovers are not a

meaningful measure of activity. What is needed is a turnover frequency. Did the authors find that reducing the catalyst loading by 1000 reduced the turnover frequency by the same factor? If not a physical explanation for why or why not is needed.

8. In responding to the original question, the authors need to give a physical explanation, not just a restatement of the observations.

Once these two issues are addressed, the work should be acceptable for publication.

Response to the Reviewers' comments:

Reviewer #2 (Remarks to the Author):

My concern has been well addressed by authors in the revised manuscript. It is now ready for acceptance.

Response: We thank the reviewer for his/her kind comments.

Reviewer #3 (Remarks to the Author):

The authors have addressed most of my questions but not all. The remaining issues are listed using the same numbering scheme as that used in the original review.

1. The authors needs to document with numbers that the cost of synthesis gas from methane or coal is lower than the price of ethylene. Simply expressing the belief is not sufficient.

Response:

We agree with the referee that expressing belief is not sufficient. However, it was not possible for us to make a reliable comparison between ethylene and CO prices on bulk scale, although we also contacted chemical producers directly. Hence, we changed the text on page 1 and added also an additional reference 10 which is concerned with this issue.

5. The authors need to give a response for why lowering the amount of catalyst by a factor of 1000 increased the number of turnovers by a factor of 500. Turnovers are not a meaningful measure of activity. What is needed is a turnover frequency. Did the authors find that reducing the catalyst loading by 1000 reduced the turnover frequency by the same factor? If not a physical explanation for why or why not is needed.

Response: We agree with the reviewer that the TON is not a meaningful measure of activity. It simply describes the stability of the catalyst system. In order to compare the activity of

the different catalyst systems we added the turnover frequency (TOF) as a new column into Table 1 ('TOF (h^{-1})'). In addition, the following statement has been added into the text on page 3. "As shown in Table 1, when reducing the catalyst loading by a factor of 1000, the activity of the catalyst system (turnover frequency; TOF) increased from 17 to 158 h^{-1} (Table 1, entry 7 *versus* 8). This surprising increase of activity is explained by the lower conversion in the latter case."

Once these two issues are addressed, the work should be acceptable for publication.

Response: We thank the reviewer for his/her kind comments.